# Oxidative Stress and Neurodegeneration: Interconnected Processes in PolyQ Diseases

**DOI:** 10.3390/antiox10091450

**Published:** 2021-09-13

**Authors:** Ioannis Gkekas, Anna Gioran, Marina Kleopatra Boziki, Nikolaos Grigoriadis, Niki Chondrogianni, Spyros Petrakis

**Affiliations:** 1Institute of Applied Biosciences/Centre for Research and Technology Hellas, 57001 Thessaloniki, Greece; gkekasioannis@certh.gr; 2Institute of Chemical Biology, National Hellenic Research Foundation, 11635 Athens, Greece; agioran@eie.gr (A.G.); nikichon@eie.gr (N.C.); 32nd Neurological Department, AHEPA University General Hospital, Aristotle University of Thessaloniki, 54636 Thessaloniki, Greece; bozikim@auth.gr (M.K.B.); grigoria@med.auth.gr (N.G.)

**Keywords:** polyglutamine, oxidative stress, proteasome impairment, microglia activation, neurodegeneration, neuroinflammation

## Abstract

Neurodegenerative polyglutamine (polyQ) disorders are caused by trinucleotide repeat expansions within the coding region of disease-causing genes. PolyQ-expanded proteins undergo conformational changes leading to the formation of protein inclusions which are associated with selective neuronal degeneration. Several lines of evidence indicate that these mutant proteins are associated with oxidative stress, proteasome impairment and microglia activation. These events may correlate with the induction of inflammation in the nervous system and disease progression. Here, we review the effect of polyQ-induced oxidative stress in cellular and animal models of polyQ diseases. Furthermore, we discuss the interplay between oxidative stress, neurodegeneration and neuroinflammation using as an example the well-known neuroinflammatory disease, Multiple Sclerosis. Finally, we review some of the pharmaceutical interventions which may delay the onset and progression of polyQ disorders by targeting disease-associated mechanisms.

## 1. Introduction

Neurodegeneration is a complex process associated with progressive degeneration of neurons and disorganization of the local neuronal circuitry. It is usually observed in the context of neurodegenerative diseases (NDDs, such as Alzheimer’s (AD) or Parkinson’s (PD) disease) and is often restricted in specific areas of the nervous system [1]. NDDs are associated with mutations in proteins which seem to be interconnected at the molecular level; for example, recent reports indicate that α-synuclein or ataxin-1 (ATXN1), proteins related to PD and spinocerebellar ataxia type-1 (SCA1), respectively, may represent risk factors for the pathogenesis and progression of AD or Multiple Sclerosis (MS) [2,3]. Revealing such interconnections may facilitate the delineation of the pathogenic mechanisms and the development of novel therapeutic approaches in NDDs.

Polyglutamine diseases (polyQ) are a group of rare, inherited and fatal NDDs caused by the expansion of trinucleotide repeats (CAG) encoding for glutamine within the coding region of various unrelated genes. There are nine known polyQ diseases including Huntington’s disease (HD), six spinocerebellar ataxias, namely SCA1, SCA2, SCA3, SCA6, SCA7 and SCA17, dentatorubral-pallidoluysian atrophy (DRPLA) and spinal and bulbar muscular atrophy (SBMA). Eight of them are inherited in an autosomal dominant manner with the exception of X-linked SBMA [4]. CAG expansions result in longer polyQ chains in the produced protein. In HD, SCA2 and SBMA, the repeats are detected in the first exon, while in SCA1, SCA7/SCA17, DRPLA and SCA3 they are located in the eighth, third, fifth and tenth exon, respectively [5,6,7,8,9,10,11,12]. CAG repeats are categorized into three main groups: normal, intermediate and pathogenic. PolyQ stretches with repeat numbers in the normal range are not associated with disease state in contrast to pathogenic ones. Intermediate repeats usually are not pathogenic but tend to increase in number causing disease in the next generations [13]. Each disease has a characteristic pathogenic threshold in polyQ expansions [14]. The number of glutamine repeats usually varies from 30 to 50 in healthy individuals but polyQ tracts in patients contain more than 40 and sometimes over 100 glutamines (Table 1) [4]. Larger polyQ tracts are associated with earlier disease onset, severe phenotype and rapid progression [15]. However, the number of polyQ expansions is not directly related to the age of the disease onset, suggesting that other genetic factors may also contribute to the pathogenesis [16].

PolyQ-expanded genes are widely expressed throughout the body; however, each disease exhibits selective neuronal degeneration [17]. The affected areas differ in each disease but there is considerable overlap, including brainstem nuclei, cerebellum, basal ganglia and spinal motor nuclei. The mechanisms for this regional selectivity remain poorly understood. Clinical symptoms in patients generally begin in midlife, although they may manifest earlier, and include progressive ataxia, atrophy and motor dysfunction [18].

Despite the fact that the genetic basis of polyQ diseases was identified, the molecular basis for their pathogenesis still remains controversial. With the exception of the polyQ-expanded region, pathogenic proteins do not share a similar structure or biological function [14]. Instead, they form characteristic inclusion bodies, a hallmark of these diseases, in the cytoplasm or nucleus of the affected neurons [19]. Aggregation of mutant proteins is associated with cytotoxicity which may result from the proteolytic cleavage of polyQ proteins into toxic fragments [20] and the formation of small protein oligomers or larger insoluble aggregates [21]. These events are polyQ length-dependent but amino acid residues outside of the polyQ stretch might be also associated with selective neurotoxicity [22]. Mutant proteins may induce cell death through loss or gain of function, transcriptional dysregulation, abnormal protein interactions with nuclear proteins (e.g., transcription factors) and dysfunction of the ubiquitin–proteasome system (UPS) [23]. Here, we review the effects of polyQ-induced oxidative stress in cellular and animal models of these diseases. We present evidence on the induction of neuroinflammation in animal models and discuss the interplay between neurodegeneration and neuroinflammation using as an example the well-known neuroinflammatory disease Multiple Sclerosis (MS). Finally, we describe compounds with a potential therapeutic value which are currently tested in preclinical or clinical studies.

## 2. Oxidative Stress

The basic concept of oxidative stress was introduced in 1985 by Helmut Sies to describe the imbalance between oxidants and antioxidants in biological systems [24]. Oxidative stress is caused by the excessive unregulated production of reactive oxygen species (ROS), such as hydroxyl radicals, hydrogen peroxide, nitric oxide and superoxide and the inability or reduced ability of biological systems to inactivate/destroy these products through direct and indirect antioxidant mechanisms [25]. It may be classified according to its intensity; low-level oxidative stress (or eustress) is necessary for redox regulation and signaling while the supraphysiological oxidative challenge is defined as oxidative distress. Oxidative distress may induce various pathological conditions including deregulated signaling pathways, impaired cellular function, damage of cellular compartments, mitochondrial dysfunction and cell apoptosis [26].

ROS are suggested to play a crucial role in oxidative stress and are generated by enzymatic or non-enzymatic processes. Enzymatic reactions include those involved in the mitochondrial respiratory chain, the cytochrome P450 system, prostaglandin synthesis and phagocytosis. Non-enzymatic generation of ROS may result from reactions of molecular oxygen with organic compounds [27]. Even though ROS are produced in several organelles, such as the endoplasmic reticulum and peroxisomes, the mitochondria are their main source [28]. ROS react with several types of molecules causing DNA, RNA and protein oxidation, enzyme inhibition, lipid peroxidation and DNA damage, eventually activating pathways of programmed cell death [29,30]. The activity of ROS may be reduced by antioxidant enzymes, such as superoxide dismutase (SOD), catalase, glutathione peroxidase and reductase and peroxiredoxins [31]. Additionally, non-enzymatic antioxidants may also decrease oxidative stress-induced damage either through direct scavenging of ROS and other free radicals, inhibition of their formation or enhancement of the expression of intracellular antioxidant enzymes [32].

The involvement of oxidative stress in the pathology of NDDs was proposed more than 20 years ago [33]. Experimental evidence indicates that oxidative stress is implicated in the progression of the most common NDDs. Aggregated proteins deposited in the brain of AD and PD patients induce the formation of free radicals and hydrogen peroxide [34]. In PD, oxidative stress contributes to the complex cascade of cellular events leading to the degeneration of dopaminergic neurons. Detailed analyses of postmortem PD brains showed increased levels of lipid peroxidation, DNA damage and RNA-protein oxidation [35]. Furthermore, the loss of dopaminergic neurons is associated with the inhibition of complex I, the first multimeric enzymatic complex of the mitochondrial electron transport chain and the subsequent increase in ROS levels [36].

SCA3 patients have increased catalase activity, higher levels of DNA damage in their peripheral blood lymphocytes and elevated ROS levels in their serum [37,38]. Evidence from postmortem samples suggests that the HD brain is characterized by enhanced levels of oxidative damage [39], including elevated 8-hydroxy-2’-deoxyguanosine (OH8dG) levels, a marker of DNA damage in the mitochondria of the parietal cortex, increased levels of carbonylated proteins and antioxidant enzymes in the striatum and cortex [40,41] and decreased numbers of striatal mitochondria [42]. Moreover, defects of the respiratory complexes and mitochondrial DNA depletion were observed in the caudate nucleus of HD patients [43] and brains of individuals suffering from infantile-onset SCA [44]. These observations point towards a heightened oxidative burden associated with polyQ diseases and indicate a potential link between their progression and mitochondrial damage induced by oxidative stress.

## 3. Oxidative Stress in Cellular and Animal Models of PolyQ Diseases

### 3.1. Overexpression Cellular Models

Cellular models of polyQ diseases provide the opportunity to study the molecular mechanisms of pathogenesis in a high-throughput and reproducible manner. These models are expected to exhibit some form of cytotoxicity and to accumulate intracellular and partially insoluble protein inclusions. Several studies describe the overexpression of exogenous mutant polyQ proteins in immortalized or primary cell lines and compare their oxidative status compared to the wild-type counterparts.

Early reports in human neuroblastoma SK-N-SH and monkey COS7 cells indicated that mutant ataxin-3 (ATXN3) with 78 glutamines decreases the reduced/oxidized glutathione ratio (GSH/GSSG), an indicator of oxidative stress and cellular health [45]. This effect was accompanied by a significant reduction in the activities of enzymatic components of the antioxidant defense, such as catalase, glutathione reductase and SOD as well as in mtDNA copy number, a biomarker associated with mitochondrial function [46]. Similarly, induced production of wild-type and mutant polyQ proteins in rat PC12 cells resulted in ROS generation, decreased activity of antioxidant enzymes and down-regulation of stress-response genes. Additionally, accumulation of ROS resulted in DNA damage and activation of ataxia-telangiectasia mutated (ATM) kinase, a DNA damage-induced protein kinase [47,48]. These results are in agreement with the ATM/ATR-dependent phosphorylation and accumulation of target proteins in fibroblasts of HD and SCA-2 patients as a response to oxidative stress-induced DNA damage [49]. Overproduction of mutant polyQ proteins is also associated with increased vulnerability to exogenous oxidative stress. HeLa cells expressing mutant *HTT* showed increased sensitivity to H_2_O_2_ and reduced viability in a polyQ-length-dependent manner. These effects were accompanied by increased nuclear localization and aggregation of polyQ proteins [50], similarly to what is observed in patient fibroblasts [51]. More importantly, oxidative stimuli led to rapid proteasomal malfunction [52]. On one hand, this may further aggravate the accumulation of misfolded proteins and on the other hand, this may further enhance oxidative damage [53].

Most of the aforementioned studies used immortalized cells which are easily maintained and expanded. One may argue that these cells have high chaperone activity while central pathways, including cell cycle regulation, are highly deregulated [54]. Nevertheless, overproduction of polyQ-expanded proteins in more relevant cellular models also reveals signs of oxidative stress. More specifically, transcript levels of 3-betahydroxysterol delta-24-reductase (DHCR24), a marker of oxidative stress and index of the cell response to oxidative stimuli greatly differ in striatal precursor cells (ST14A) expressing *HTT* with a normal number of repeats (Q15) compared to cells expressing *HTT* with pathogenic (Q120) polyQ tracts [55]. Furthermore, the effect of mutant *HTT* on mitochondrial dynamics and oxidative stress was studied in the STHdh striatal cellular model of HD. The activity of the nuclear factor erythroid 2-related factor 2 (Nrf2), the main regulator of cellular antioxidant response [56], was reduced in StHdh*^Q111/Q111^* cells and was accompanied by reduced expression of key regulators of the Nrf2 signaling. Additionally, StHdh*^Q111/Q111^* cells contained more swollen and fragmented mitochondria with a higher oxidative state after treatment with H_2_O_2_ compared to StHdh^Q7/Q7^ cells [57], indicating that cells producing the polyQ-expanded protein undergo mitochondrial stress possibly due to oxidative damage since H_2_O_2_-dependence was revealed.

Our group has recently generated a novel inducible protein aggregation model of polyQ-expanded ATXN1 in human mesenchymal stem cells (MSCs). Gradual formation of insoluble ATXN1(Q82) protein inclusions in MSCs was accompanied by ROS accumulation and deregulation of pathways involved in oxidative phosphorylation, which is also observed in the cerebellum of SCA1 patients [58]. More importantly, a proteomics analysis indicated that aggregation of the polyQ-expanded protein was associated with the deregulation of sodium ion transport and the mitochondrial respiratory chain, suggesting a cellular deficiency in ATP synthesis and consequently in energy metabolism [59].

### 3.2. Induced Pluripotent Stem Cell Models

Induced pluripotent stem cells (iPSCs) are excellent tools for modeling neurological diseases and for the identification of pathological molecular mechanisms in iPSC-derived neurons [60]. Various iPSC lines were developed to study the pathogenesis and pathophysiology of polyQ diseases. Nevertheless, the impact of oxidative stress and DNA damage in iPSC-derived neurons is still unclear and requires further investigation.

Chuang et al. studied the role of glutamate-induced neurotoxicity in polyQ disease pathology. Using iPSCs from SCA2 and SCA3 patients, they investigated the effect of glutamate on mitochondrial function. According to their results, SCA-iPSCs were more vulnerable to glutamate-induced mitochondrial stress than normal iPSCs [61]. A complementary analysis using normal human embryonic stem cells (hESCs), control iPSCs and iPSCs from HD patients identified 26 genes that were significantly deregulated in all patient cell lines and were implicated in oxidative stress response. Moreover, the levels of oxidative stress-related proteins, such as SOD-1 and peroxiredoxin were significantly up-regulated in HD-iPSCs, suggesting that these cells have an active oxidative stress response [62]. Transcriptional alterations were also detected in iPSC lines derived from two HD patients with a different number of glutamine repeats (71 and 109 polyQ repeats). Among the analyzed genes, transcripts involved in DNA damage response were significantly deregulated in HD109 iPSCs [63].

A more systematic analysis was conducted by the HD iPSC Consortium. Fourteen patient and control iPSC lines were generated to model HD in iPSC-derived neurons. HD neural precursor cells (NPCs) exhibited significantly lower levels of ATP suggesting a compromised energy metabolism while differentiated HD cells were more vulnerable to H_2_O_2_-induced oxidative stress which led to enhanced rates of cell death compared to control iPSCs [64]. Assays in a live-cell metabolic platform using iPSC-derived neurons indicated impairment of glycolytic enzymes and deficits mainly in the glycolytic capacity in a polyQ-length-dependent manner while oxidative phosphorylation was minimally affected in HD neurons [65].

Despite the effort, it is not fully understood how polyQ-expanded proteins can be the cause of increased oxidative burden. Mutant proteins may interfere with numerous cellular processes which collectively lead to ROS production [66]. Toxic polyQ fragments associate with mitochondria and reduce their transport rate and their membrane potential [67,68]. PolyQ inclusions may also interfere directly with mitochondrial membranes and inhibit complexes of the electron transport chain disrupting mitochondrial function [69]. Furthermore, a study in ESCs of HD mice indicated that oxidative stress increases the number of CAG triplet expansions and this event is affected by the DNA repair efficiency of differentiated cells [70]. Whether oxidative stress may lead to genomic instability through DNA damage and defects in the DNA-repair machinery [13] remains to be investigated (Figure 1).

### 3.3. Animal Models of PolyQ Diseases

Rodent and invertebrate HD models were extensively studied to reveal whether there is causality between disease progression and oxidative stress and, by extension, whether ameliorating oxidative stress is a reasonable therapeutic approach. Multiple studies have reported increased oxidative damage in R6/1 mice expressing exon 1 of human *HTT* with 116 CAG repeats under the human HTT promoter [71]. Higher levels of lipid peroxidation and increased activity of the antioxidant enzyme SOD are detected in these mice even when the neurological phenotypes are still rather mild [72]. In older animals SOD activity declines, possibly indicating a failure of the antioxidant defenses as the disease progresses and the neurological phenotypes become more severe. It was suggested that nitric oxide (NO) that forms the highly reactive metabolite peroxynitrite is responsible for oxidative damage in R6/1 mice. The expression and activity levels of nitric oxide synthase (NOS) are elevated in 19-week-old R6/1 mice suggesting an association with the onset of the progressive neurological phenotype manifested by these animals [73].

Evidence of raised oxidative burden was also found in R6/2 mice expressing exon 1 of human *HTT* with 144 CAG repeats [71]. Increased oxidative damage of nuclear and mitochondrial DNA in the striatum of R6/2 mice and enhanced levels of carbonylated proteins are a few of the markers manifested by this mouse model [74,75,76]. Proteomic analyses have revealed differentially produced proteins involved in oxidative stress defense, oxidative phosphorylation, TCA cycle and pyruvate metabolism suggesting mitochondrial dysfunction [77]. Moreover, increased levels of the antioxidant glutathione are detected in cortical and striatal mitochondria of R6/2 mice [78], possibly suggesting a cellular effort to protect mitochondria before they become dysfunctional.

On the contrary, data on the presence of oxidative stress in YAC128 mice carrying a yeast artificial chromosome encoding for the entire human *HTT* gene with 128 CAG repeats [79] are inconsistent. Although a proteomic analysis revealed differentially expressed genes involved in oxidative stress defense and mitochondrial metabolism, isolated mitochondria from the striatum of YAC128 mice showed no differences in respiration and Ca^2+^ uptake capacity compared to mitochondria from wild type mice, thus weakening the possibility of a mitochondrial defect [77,80]. Moreover, even at 12 months of age, YAC128 mice do not show increased lipid peroxidation or elevated levels of carbonylated proteins [81]. Nevertheless, one study reports an abnormally increased pool of glutathione in the striatum of YAC128 mice thus suggesting an elevated oxidized state [82]. Although YAC128 mice do not seem to have an excessive oxidative burden as opposed to R6/1 and R6/2 mice, the occurrence of their depressive-like behaviors can be reduced upon chronic treatment with the antioxidant probucol [83]. Overall, although YAC128 mice exhibit a large number of HD-related phenotypes, they do not manifest oxidative damage, as opposed to what was reported in human patients.

It is not yet fully understood how mutant *HTT* may cause increased oxidative burden in vivo. Reduced activities for respiratory complexes II and IV were reported in the striatum and cerebral cortex of R6/2 mice [84,85]. Impaired ascorbic acid transport from glia to neuronal cells is another factor possibly contributing to the extensive oxidative damage associated with mutant *HTT* [86,87,88]. This defect was shown to precede neurological phenotypes in R6/2 mice and it is considered to deprive neurons of an important antioxidant defense.

Only a handful of evidence exists on the involvement of oxidative stress in SCA1 in animal models. Proteome profiling of the heterozygous ATXN1(Q154) mouse [89] at the symptomatic stage, revealed changes in mitochondrial proteins that could lead to elevated oxidative stress. Treatment of both symptomatic and pre-symptomatic animals with the mitochondria-targeted antioxidant MitoQ delayed the progressive loss of motor coordination and prevented oxidative stress-induced DNA damage [90]. One study in *Drosophila* revealed that the expression of Lazarillo-related lipocalins, a family of proteins that exerts neuroprotective action upon oxidative stress, can prevent neurodegeneration in a SCA1 fly model [91]. This suggests that oxidative damage might contribute to neurodegeneration although the overproduction of those proteins is a rather indirect proof. Overall, the impact of oxidative stress in SCA1 has not yet been extensively studied in animal models.

In contrast, oxidative stress may play a more central role in the SCA3 pathology. Evidence from animal models suggests that antioxidants may have a beneficial effect on the progression of the disease. Specifically, creatine treatment of mice expressing *ATXN3(Q135)* led to an improvement in motor performance. Although creatine is not an antioxidant per se, it improves mitochondrial function while in the above-mentioned mice it also reduced the production of antioxidant enzymes [92]. Flies expressing *ATXN3(Q78)* were also found to have increased levels of oxidized proteins while overproduction of the antioxidant cystathionine γ-lyase reduced the oxidative burden of these flies and protected them from degeneration [93]. In the same SCA3 *Drosophila* model, caffeic acid and resveratrol increased the lifespan and ameliorated locomotory defects by enhancing the transcriptional activity of the Nrf2 transcription factor [94,95]. Taken together, these studies suggest that increased oxidative stress may, at least partially, underlie SCA3 pathology.

## 4. Molecular and Cellular Effects of Oxidative Stress in PolyQ Diseases

### 4.1. Proteasome Modulation

The proteasome is a large multi-subunit enzymatic complex that catalyzes the degradation of both normal and abnormal proteins in an ATP-dependent or independent manner [96]. It has three distinct proteolytic activities: the chymotrypsin-like (CT-L), the trypsin-like (T-L) and the peptidyl-glutamyl-peptide hydrolyzing or caspase-like (C-L) [97]. It consists of the main complex called 20S core while binding of the 19S regulatory multi-subunit complex, is possible either at one (19S:20S, known as 26S proteasome) or both edges (19S:20S:19S, referred to as 30S proteasome complex) of the 20S core. In the ATP-dependent degradation, the protein to be degraded is recognized by the proteasome through tagging with a small, evolutionarily conserved protein, namely ubiquitin; at least four molecules of ubiquitin serve as a signal for proteasomal degradation. This conjugation occurs through the tightly coordinated action of three different enzymes, namely E1, E2 and E3 [98]. Peptides and partially unfolded proteins, including oxidized proteins, were reported to be degraded in an ATP- and ubiquitin-independent manner [99]. A large amount of data show that proteasome-mediated protein degradation is enhanced upon exposure to oxidants. Moreover, increased oxidative damage can interfere with the function of the proteasome itself [100]. On this basis, it is not surprising that the proteasome is impaired in HD patient brains and implicated in polyQ pathogenesis.

Reduced CT-L proteasome activity was reported in the brains of HD patients [101] which may be partially due to the sequestration of proteasome subunits into insoluble polyQ protein inclusions [58]. Data from yeast and cell lines containing polyQ-expanded proteins clearly show proteasome inhibition [102,103] while overexpression of proteasome subunits rescues the cytotoxic phenotypes of mutant proteins [104]. Nonetheless, evidence from animal models on the activity and involvement of the proteasome in polyQ diseases is conflicting. For instance, proteasome activity remained unchanged in the brain of R6/1 and R6/2 mice [105,106] but another study reported increased proteasome activity in R6/2 mice [107]. On the contrary, CT-L activity was reduced in YAC128 mice [108]. It is possible that these differences may be attributed to different stages of the disease in each animal model.

Data from *C. elegans* support the findings from rodent HD models. More specifically, the accumulation of ubiquitin in the neurons of a nematode strain producing polyQ aggregates indicates UPS blockage [109]. Enhancement of the proteasome activity through the overexpression of one of its subunits (*rpn-6* or *pbs-5*) was shown to delay the paralysis of nematodes expressing polyQ in the body wall muscle cells and to improve locomotory defects in nematodes expressing polyQ in neurons [110,111]. Importantly, a genome-wide RNA interference screen in nematodes identified seven subunits of the 20S core, four putative subunits of the 19S cap, one ubiquitin-like protein and various enzymes involved in ubiquitin activation and deubiquitination to be negative effectors of polyQ aggregation [112].

Despite the contradictory results, it is quite clear that the proteasome plays a role in the progression of the disease. Whether proteasome activation assists in the clearance of oxidized proteins that accumulate in HD or it degrades mutant Htt before it gets highly aggregated, or both is not completely clear yet. Regardless of its exact role, the proteasome comprises a promising therapeutic target against HD as it is for many other aggregation-related NDDs [98].

Several lines of evidence from SCA1 cellular and animal models and patients reveal the role of the proteasome in SCA1 pathology. Inclusions of mutant ATXN1 stain positively for the 20S proteasome in neurons from transgenic mice and SCA1 patients [113] whereas experiments in cells have shown that ATXN1[82Q] is a proteasome substrate [114]. Moreover, a non-coding RNA (ncRNA) profiling of the cerebellum and cortex of SCA1 patients revealed that the primary targets of the enriched ncRNAs were UPS components [115]. Data from *Drosophila* coincide with the delineated role of the proteasome in ATXN1 turnover since proteasome inhibition in transgenic flies expressing the mutant isoform resulted in its accumulation [116].

The role of the proteasome in SCA3 is also very central. This is mostly due to the fact that ATXN3 has ubiquitin-interacting motifs (UIM) thus acting as a deubiquitinase (DUB) while it was also proposed to regulate the aggresomal formation of mutant proteins (mutant SOD1) through preferential cleavage of K63-linked polyubiquitin chains [117,118]. In support of this finding, the 26S proteasome was found to co-localize with polyQ aggregates in tissue from SCA3 patients [119]. In transgenic mice expressing *ATXN3(Q69)*, overexpression of the GTPase CRAG that promotes ubiquitination and degradation of polyQ aggregates resulted in extensive polyQ aggregates clearance and re-activated dendritic differentiation in the Purkinje cells [120]. More direct evidence of the beneficial effect of the enhanced proteasome-mediated degradation of ATXN3 in vivo comes from a study in *Drosophila*; overproduction of the ubiquitin chain assembly factor E4B resulted in suppressed neurodegeneration [121].

### 4.2. Autophagy Modulation

The term autophagy describes the lysosome-dependent degradation of cytoplasmic components, a process highly conserved in all eukaryotes [122]. It is usually responsible for the bulky degradation of cytosolic content as well as for the selective clearance of aggregated proteins. Although there are suggestions that the proteasome has a much greater role in the clearance of polyQ-containing proteins [123], the active involvement of autophagy was shown in various cellular and animal models. This is rather expected given that autophagy is activated in response to oxidative stress as a protective mechanism and that multiple lines of evidence suggest that its inhibition leads to the accumulation of oxidative damage [124,125].

Htt represents the most studied polyQ protein in the context of autophagy. Experimental evidence indicates that Htt plays a physiological role in this process. More specifically, a study in yeast showed that Htt shares interacting partners and structural similarities with proteins encoded by the autophagy-related genes [126]. Htt localizes in the membranes of the ER and translocates into the nucleus upon ER stress. Once the stress is relieved, the protein may relocate back to the ER. Nevertheless, a polyQ stretch may prevent Htt relocation, thus causing ER perturbations and an increase in the number of autophagic vesicles [127]. Evidence from cells derived from a mouse HD model and lymphoblasts from HD patients indicates that mutant Htt may impair the cellular ability to degrade cytosolic components. This is mainly due to defects of cargo loading in the autophagosomes which finally fuse with the lysosomes [128]. Therefore, polyQ expansions in Htt may greatly affect the process of autophagic cargo turnover.

Modulation of autophagy was shown to affect phenotypes associated with polyQ-expanded Htt in several animal models. Intuitively, one would expect that activation of autophagy would counteract polyQ-induced toxicity, while inactivation of autophagy would induce additional detrimental effects. This is indeed supported by evidence from invertebrate models of HD. In *C. elegans*, genetic inactivation of autophagy genes exacerbates polyQ40-induced muscle dysfunction and enhances the degeneration of nematode neurons expressing human *HTT(Q150)* [128]. Although the relation between polyQ-induced toxicity and autophagy seems to be straightforward in invertebrate models, it is rather complicated in mammals. More specifically, both enhancement and inhibition of autophagy were shown to improve brain pathology and to ameliorate disease phenotypes in HD mouse models [129,130].

Although less extensively studied, the role of autophagy was also studied in SCA1 and SCA3. Regarding SCA1, it is not clear yet whether autophagy would represent a promising therapeutic target. This might be due to the presence of a nuclear localization signal that is responsible for the translocation of ATXN1 into the nucleus and protects it from autophagy-mediated degradation [131]. In fact, evidence from mammalian cells shows that cytosolic ATXN1 inclusions are stained for autophagic markers, while the nuclear inclusions are not [132]. Concerning SCA3, experimental data suggest that autophagy might be a potential therapeutic target. Similar to Htt, ATXN3 seems to play a physiological role in autophagy. Evidence from mammalian cells and nematodes indicates that ATXN3 stimulates autophagic degradation early in the autophagic pathway [133]. Unlike HD models, SCA3 models indicate a uniform behavior concerning the role of autophagy in SCA3-related phenotypes. For example, loss of p62, a regulator of selective autophagy, in a SCA3 *Drosophila* model resulted in exacerbation of eye degeneration [134]. Findings from mammalian models seem to be in agreement with those observed in invertebrates. More specifically, overexpression of beclin-1 (an autophagy enhancer) in a rat model of SCA3 led to mutant ATXN3 clearance and neuroprotection [135].

### 4.3. Mitophagy Suppression

The term mitophagy was first introduced in 2005 to describe the selective degradation of mitochondria through autophagy [136]. Precise regulation of this pathway is important for the maintenance of cellular homeostasis. Deficient removal of damaged mitochondria as the ones observed in polyQ diseases is associated with the progression of age-related NDDs. Therefore, mitophagy impairment represents one of the key components of neuronal degeneration [137]. while it may also increase ROS levels and augment oxidative stress.

Most of our knowledge regarding the involvement of mitophagy in polyQ diseases comes from HD models. Experimental evidence indicates that Htt: (1) acts as a scaffold protein, stabilizing protein complexes during the initiation of mitophagy and (2) interacts with mitophagy receptors promoting the degradation of damaged mitochondria. Therefore, polyQ-expansions in Htt inhibit its physiological function and impair mitophagy [138]. Furthermore, mutant HTT selectively interacts with mitochondrial GAPDH leading to its inactivation. Inactivated GAPDH blocks the sequestration of damaged mitochondria into the autophagosomes and their degradation into the lysosomes [139]. In total, these processes lead to the accumulation of damaged mitochondria, increased ROS levels and eventually to cell death (Figure 1).

### 4.4. Microglia Activation

Microglia are a specialized form of macrophage cells acting as the main form of immune defense within the central nervous system. Under normal, resting and surveillance conditions, microglial cells display a ramified morphology. Several stimuli such as infection, trauma, ischemia, neurodegeneration or altered neuronal activity can activate microglia. When activated, microglia assume a larger amoeboid-like shape while their function and gene expression are profoundly altered [140]. Microglial cells are the primary mediators of neuroinflammation since they secrete various inflammatory molecules; their activation was mainly blamed for the neuronal dysfunction and death involved in NDDs [141]. Activated microglia or “reactive microglia” is found in the cortex and striatum of HD patients [142]. Importantly, the level of microglial activation seems to correlate with the severity of the disease [143].

Cultured microglial cells become active following treatment with lipopolysaccharide (LPS) and various cytokines [144]. This activation is accompanied by proteasome activation as well as by the sudden massive production of free radicals, a response also known as “oxidative burst”. These free radicals are mainly produced by NADPH oxidase that is also stimulated upon microglia activation [145]. ROS production by the reactive microglia can have harmful effects on neurons therefore, reactive microgliosis could be an additional factor contributing to the elevated oxidative burden found in the HD brain [146].

Reactive microglia and its involvement in HD pathology were extensively studied mainly in HD mouse models. Although amoeboid-shaped microglia were not detected in the striatum of R6/1 mice, PSD-95-positive puncta inside microglial cells indicate synaptic pruning that precedes symptom manifestation [147]. Moreover, the interaction between neurons and microglia that prevents the latter from becoming reactive, is affected in these mice [148]. On the other hand, microglia activation in R6/2 mice seems to be more pronounced. In 13-week-old R6/2 mice, microglia were intensely stained for Iba1 (Ionized calcium-binding adaptor molecule I) indicating its activation [149]. Finally, YAC128 seems to have the weakest microglia involvement since only one study reports a significantly altered microglia morphology albeit without indications of activation [150]. Interestingly, the lack of oxidative damage in YAC128 mice is accompanied by the absence of clear signs of microglia activation.

Microglia activation may contribute to the progression of both SCA1 and SCA3 but, at the moment, the relevant evidence is very limited [151]. More specifically, experimental evidence from mouse models suggests that microglial activation may be strongly implicated in the progression of SCA1. In a SCA1 mouse model, astrocytes and microglia were found to be activated at an early time-point even when expression of mutant *ATXN1* was limited to the Purkinje cells [152]. Chemical depletion of microglia at an early stage of the disease was shown to ameliorate motor deficits in mice expressing *ATXN1(Q82)* [153]. A slight increase in microglia was also detected in SCA3 patient brains [154]; nevertheless, microglia activation has not been extensively studied in SCA3 animal models.

## 5. Oxidative Stress, Neurodegeneration and Neuroinflammation

Neuroinflammation is a complex innate immune response occurring in the nervous system [155]. It is mostly mediated by glial cells which are highly sensitive to any stimulus that may signal deviation from the healthy microenvironment [156]. Glial cells are a heterogeneous group consisting of microglia, astrocytes and oligodendrocytes. The complex interplay of these cell types shapes a variety of responses dependent not only on the type of the initial stimulus but also on the mechanisms of glial cell activation. Moreover, glial cells maintain a dual role as mediators of inflammation: their activation phenotype may be characterized as pro- or anti-inflammatory: M1/M2 and A1/A2 for microglia and astrocytes, respectively [157,158]. Interestingly, these activation phenotypes are not considered as polar opposites but rather, as broad categories of several activation subtypes [159]. As a response to potentially damaging stimuli, microglia and astrocytes interact regulating the pro-/anti-inflammatory effect towards neurons [160]. Moreover, continuous pro-inflammatory stimulation may lead to chronic, self-sustained neuroinflammation which evidently accompanies NDDs, especially of the immune-mediated type, such as MS.

As previously mentioned, neuroinflammation may be induced by a variety of factors, including brain injury, toxic metabolites or oligomeric protein species [161]. Could the events described in the previous sections trigger the development of acute or chronic neuroinflammation in the polyQ brain? The effect of neuroinflammation in the progression of polyQ diseases remains largely unknown but it is mainly suggested by indirect experimental evidence. PolyQ proteins are ubiquitously expressed in neuronal and non-neuronal cells [162]; their mutant isoforms may impair the function of glial cells [163] which regulate neuronal signaling and protect neurons from oxidative damage [164]. Additionally, microglia expressing mutant *HTT* have increased levels of pro-inflammatory cytokines [165] which may induce the activation of caspases and the production of free radicals. If released in the brain, these cytokines may also disrupt the blood–brain barrier allowing infiltration of immune cells into the brain. Furthermore, immune cells producing polyQ-expanded proteins show clear signs of activation of the peripheral immune system [165,166]. These observations may reflect neuroinflammatory events occurring in the central nervous system (CNS) and could be used for the pre-symptomatic diagnosis of polyQ diseases [167].

Interestingly, recent studies suggest the association between neuroinflammation and neurodegeneration; *ATXN1*, the causative gene of SCA1 was recently identified as a susceptibility locus for MS [3]. Moreover, a role for polyQ-expanded ATXN1 was described in the Pathogenesis of Experimental Autoimmune Encephalomyelitis (EAE), the rodent model for MS [168]. As the mutant protein primarily caused deregulation of the B cell activity, this work highlights a previously underestimated role of ATXN1 in immune-mediated neurodegeneration and its implication in MS. MS is a neuroinflammatory demyelinating disease of the CNS [169] and has recently emerged as a disease with a strong neurodegenerative component [170]. The co-occurrence of neuroinflammation with neurodegeneration, possibly at all stages of the disease, is further supported by dynamic visualization models that provide a unifying approach to the relative contribution of the two pathological processes. In these models, neurodegeneration is considered as the main process underlying the loss of the baseline functional capacity of the CNS, while neuroinflammation is considered a constantly recurring process [171].

Oxidative stress is evident in the brain of patients with MS and is associated with an imbalance between ROS generation and the activity of antioxidant mechanisms [169]. It affects neurons and glia cells particularly in the context of the chronic inflammatory process [172], which characterizes the “smouldering” MS lesions [173]. Activated microglia and macrophages exhibit an inherent capacity to produce reactive oxygen and nitric oxide species [174]. Inflammatory damage causes injury and chronic demyelination in axons in axons of the white and grey matter [170]. Moreover, degenerating neurons release iron into the extracellular space, which further contributes to oxidative stress [175,176]. Mitochondrial dysfunction and reduced ATP levels render neurons, susceptible to cellular dysfunction and, therefore, to neurodegeneration [177]. This combination of oxidative stress and mitochondrial dysfunction disrupts cellular communication even in the absence of actual cell loss [172]. Furthermore, mitochondrial DNA mutations within patients’ neurons are possibly caused by inflammatory episodes rendering neurons particularly susceptible to oxidative stress [178] (Figure 2).

These studies underline the complex interplay between oxidative stress-induced neuroinflammation and neurodegeneration in MS, a disease traditionally considered as of autoimmune inflammatory origin in the CNS. It has recently become evident that neuroinflammation in MS is associated with age and with various degrees of neurodegeneration. In this frame, oxidative stress has emerged as a mechanism of tissue damage linked with both chronic inflammation and neurodegeneration. This mechanism indicates potential targets of pharmaceutical intervention which may be relevant also for patients suffering from polyQ diseases.

## 6. Pharmacological Strategies against PolyQ Diseases

At the moment, there is no cure for polyQ diseases. Antisense oligonucleotides specifically targeting the polyQ-expanded allele and aiming in reducing the levels of the aggregation-prone mutant protein are currently tested in preclinical and clinical studies with promising results [179]. Complementary to this approach, treatment of patients with antioxidants or compounds targeting the degradation machineries or microglia (Figure 3), might mitigate the symptoms and the progression of the disease.

### 6.1. Preclinical Studies

#### 6.1.1. Antioxidants

Regardless of the factors contributing to the buildup of oxidative burden, it was repeatedly shown that antioxidants attenuate several of the phenotypes manifested in polyQ animal models. For instance, tolfenamic acid improved motor coordination and memory in R6/1 mice while it also enhanced the mRNA levels of glutathione and Nrf2 [180]. In the same mouse model, N-acetylcysteine (NAC) delayed the onset and progression of motor defects while it rescued the mitochondrial respiratory deficit occurring in the striatum and cortex. NAC treatment also reduced the levels of carbonylated proteins present in the striatal mitochondria of the R6/1 [181]. Antioxidants were also shown to rescue multiple HD-associated phenotypes in R6/2 mice. The antioxidant BN82451 improved motor performance, reduced brain and neuronal atrophy while it extended the lifespan of R6/2 mice [181]. The survival of R6/2 mice was also extended through co-administration of Coenzyme Q10 and creatine while motor performance was also improved [182]. Nordihydroguaiaretic acid (NDGA), an antioxidant that inhibits lipid peroxidation, restored mitochondrial structure, membrane potential and synapse structure in the striatum and increased the lifespan of R6/2 mice [183]. Finally, neutralization of reactive radical species directly in the mitochondria can also effectively attenuate oxidative stress in HD models. The synthetic antioxidant XJB-5-131 was shown to attenuate or reverse the disease progression when administered to HdhQ150 mice. Specifically, it promoted weight gain, prevented neuronal death, reduced mtDNA damage in neurons and slowed down the decline of motor performance [184].

Although only a few reports on the oxidative burden of invertebrate polyQ models exist, several antioxidants were tested in *C. elegans* and *Drosophila* mostly for their anti-aggregation properties and their effectiveness in rescuing phenotypes induced by polyQ-overexpression. Treatment of transgenic nematodes producing a Q40 protein with the antioxidants Vitamin C, α-lipoic acid and epigallocatechin failed to ameliorate movement defects and polyQ aggregation [185]. However, when the same nematode model was treated with the antioxidant phycocyanin the number of polyQ aggregates was reduced, suggesting that all antioxidants are not equally effective [186]. Several other antioxidants were reported to have beneficial effects in nematode HD models. Diphenyl diselenide reduced ROS levels and extended the lifespan and healthspan of both wild type and polyQ transgenic worms while it also reduced polyQ aggregation in the muscle and the polyQ-mediated death of ASH neurons [187]. Similarly, both the guarana hydroalcoholic extract and the water extract from roasted guarana seeds reduced the number of polyQ aggregates in the muscle of nematodes [188,189]. The former also reduced polyQ-mediated neuronal death of ASH sensory neurons [188]. In nematodes expressing *ATXN3(Q130)*, treatment with the antioxidants rapeseed pomace, *Hyptis suaveolens, Hyptis pectinata* and *Hyptis marrubioides* extracts improved their motility [190,191].

In *Drosophila* flies expressing polyQ-expanded genes, treatment with several antioxidants such as fisetin, resveratrol, grape seed polyphenolic extract, curcumin and caffeic acid, had various beneficial effects such as reduced ROS production, neuroprotection, lifespan extension and improvement of polyQ-induced motor dysfunction [95,192,193,194]. Notably, it was proposed that antioxidants-mediated ROS neutralization may hamper autophagy that is normally triggered by ROS. More specifically, it was shown in *Drosophila* that high concentrations of NAC could block autophagy and exacerbate the HD phenotype while they could also reverse the benefits of autophagy-inducing agents such as rapamycin [195]. Therefore, the antioxidant properties of a compound, the applied concentration and the duration of the treatment should be carefully considered when it comes to a specific treatment.

#### 6.1.2. Proteasome Activators

Despite the contradictory results, many studies agree that proteasome activation through pharmacological or genetic manipulation has beneficial effects in rodent and invertebrate polyQ models. For example, benzamil, an agent used to rescue acid-sensing ion channel (ASIC)-dependent cytotoxicity, resulted in increased proteasome activities in R6/2 mice and thus in the enhanced degradation of soluble Htt. Moreover, motor deficits were attenuated and lifespan was prolonged [196]. Another compound boosting proteasomal activity, namely baclofen (a GABAB receptor agonist), was found to improve motor coordination in YAC128 mice [108]. Interestingly, some evidence indicates that mutant Htt may lead to reduced histone acetylation through the interaction with specific transcription factors. Treatment of the N171-Q82 (N-terminal fragment incorporating both exon 1 and exon 2 of *HTT* gene with 82 CAG repeats) HD mouse model with a histone deacetylase inhibitor resulted in the upregulation of the RNA expression levels of various UPS components. Treatment of these mice with this inhibitor also promoted lifespan extension and attenuation of the neurological phenotypes [197].

#### 6.1.3. Autophagy Inducers

More than a decade of research has shown that autophagy activation may be a potential therapeutic strategy for polyQ diseases. For example, treatment of transgenic flies expressing human *HTT*(Q128) with an autophagy-enhancing small molecule, AUTEN-67, improved their climbing ability and moderately extended their lifespan [198]. Similarly, chemical modulation of a transcriptional activator of autophagy using chenodiol, fenbufen and sulfaphenazole rescued the motor-deficient phenotypes manifested in transgenic ATXN3 nematodes [199].

Rapamycin, a lipophilic macrolide antibiotic induces autophagy by inhibiting the activity of mTOR; its administration in mice and cells reduced the toxic effects of polyQ-expanded Htt. Similarly, administration of CCI-779 (cell cycle inhibitor-779, temsirolimus), a rapamycin ester with more desirable pharmaceutical properties compared to rapamycin, improved the performance of an HD mouse model in rotarod and grip strength and reduced its tremors [200]. CCI-779 also reduced the number of ATXN3-positive brain aggregates and improved motor performance in a SCA3 mouse model [201]. Accordingly, cordycepin, a compound that induces autophagy, reduced both aggregated and soluble ATXN3 levels in a lentivirus-based mouse model of SCA3 and improved the neuropathology of cerebellar Purkinje neurons and the motor deficits manifested by these mice [202].

The antioxidant molecule resveratrol prevented ROS generation in dopaminergic neurons producing mutant Htt and facilitated its autophagic degradation by rescuing autophagosome formation [203]. Another study showed that trehalose, an mTOR-independent inducer of autophagy, inhibited the formation of mHTT aggregates and restored the motor function in an HD mouse model [204]. Likewise, the mTOR-independent small molecules SMER10, SMER18 and SMER28 reduced polyQ-induced toxicity enhancing the clearance of autophagic substrates in cells and *Drosophila* [205]. Furthermore, a screening of FDA-approved drugs for autophagy enhancers identified novel autophagy-inducing compounds. Loperamide, nimodipine, minoxidil and rilmenidine promoted autophagy in cells and reduced mutant Htt aggregation and cytotoxicity [206]. Such autophagy-inducing compounds may also provide the basis for the discovery of mitophagy inducers.

#### 6.1.4. Microglia Suppressors

Several studies have shown that reduction of microglia activation in HD mouse models may confer a positive outcome on the severity of neurological symptoms. Treatment of R6/2 mice with INO-1001 and Olaparib, which are PARP-1 (poly-ADP ribose polymerase) inhibitors, led to reduced microglial activation, attenuation of neurological phenotypes and lifespan extension [207,208]. The same benefits were recorded following treatment with doxycycline in R6/2 mice [209]. Finally, conditioned medium from mesenchymal/stromal cells of the amniotic membrane (known to confer protection in vitro and in vivo in animal models of immune-based disorders and of traumatic brain injury) was shown to ameliorate HD-associated symptoms and to extend the lifespan of R6/2 mice while at the same time it reduced reactive microgliosis [210]. Although the exact molecular mechanisms underlying the effects of these interventions on microglia activation are not clear yet, it seems that reduction of microglia activation is a potentially promising therapeutic venue.

### 6.2. Clinical Trials Using Antioxidants

A variety of pharmaceutical compounds with presumed anti-oxidant or anti-inflammatory activity is currently under investigation and few of them exhibit promising results in pilot and/or phase 1/2 trials. For example, treatment of HD patients with d-α-tocopherol had a beneficial effect on neurologic symptoms when it was administered early in the course of the disease [211]. Two smaller trials tested the efficacy of highly unsaturated fatty acids (HUFAs) and ethyl-ester of eicosapentaenoic acid (ethyl-EPA) in HD patients and revealed motor improvement [212,213]. More double-blind, placebo-controlled studies with large patient cohorts are needed to define the exact antioxidant compounds, the disease state where treatment should begin and the duration of the treatment that may slow down or reverse the progression of polyQ diseases.

More evidence is obtained from clinical trials in MS patients. Statins, inhibitors of 3-hydroxy-3-methylglutaryl coenzyme A reductase, were previously shown to exert immunomodulatory, anti-inflammatory and anti-oxidant effects [214,215], have long been advocated as potential therapeutics for MS [216]. Lipoic acid is an endogenously produced small molecule with an anti-oxidant effect as it induces free-radical scavenging activity, metal ion chelation, regeneration of glutathione, and repair of oxidative damage [217]. These functions appear of particular significance in relation to mitochondria maintenance and the associated oxidative respiration [217]. Lipoic acid assists in the maintenance of vascular endothelial integrity and exert a role in Nrf2 activation [218]. In a single-center double-blind, randomized trial, 1200 mg lipoic acid (*n* = 22) vs. placebo (*n* = 24) were administered orally over a period of 2 years. Patients who received lipoic acid exhibited 68% reduction in annualized percent decrease in the brain volume suggesting a clinical benefit [219].

Recently, Bruton Tyrosine Kinase (BTK) inhibitors, agents with a capacity to penetrate the brain parenchyma, have emerged as promising tools that may target chronic inflammation [220]. BTK, a tyrosine kinase1, participates in the signaling of activated myeloid cells present in the CNS and surrounds the chronic active, “smouldering” lesions of MS (macrophages and microglial cells). In this respect, BTK inhibitors may have a role in blocking the activation of microglia and the associated pathological cascade mediated by microglia-dependent pro-inflammatory milieu and oxidative stress [220].

## 7. Conclusions

The experimental evidence reviewed here argues the direct involvement of oxidative stress in the neurodegeneration observed in the brain of patients suffering from polyQ diseases. It also suggests that the induction of neuroinflammation—acute or chronic—may partially contribute to the gradual clinical deterioration characterizing the disease; its pharmacological inhibition may mitigate the progression of polyQ diseases.

## Figures and Tables

**Figure 1 antioxidants-10-01450-f001:**
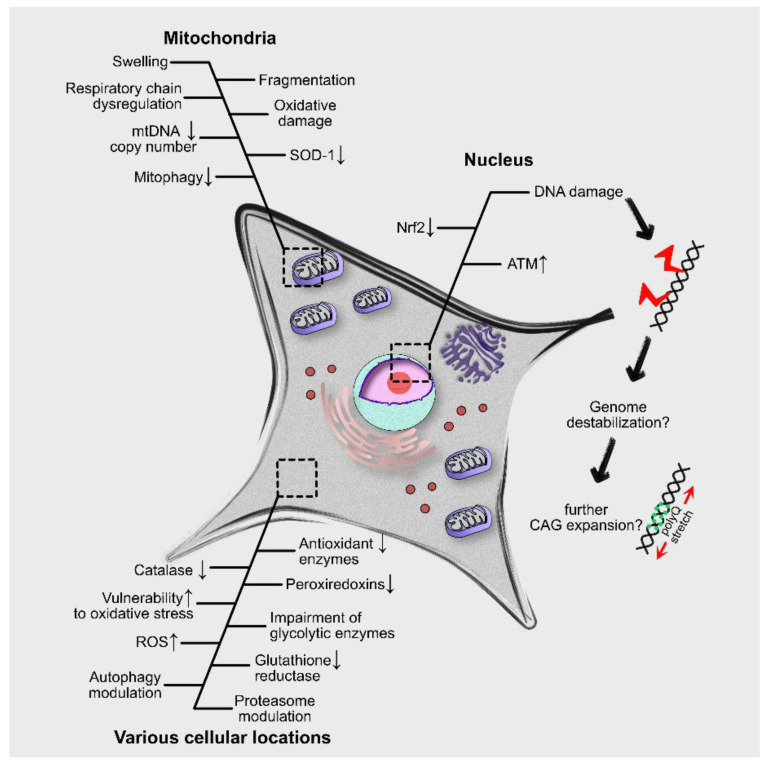
Effects of polyQ-expanded proteins at the cellular level. PolyQ-expanded proteins induce responses from multiple cellular components including the mitochondria and the nucleus. Accumulation of mutant proteins may lead to excessive oxidative stress, mitochondrial damage, modulation of autophagy and proteasome activity and, possibly, genomic destabilization through DNA damage.

**Figure 2 antioxidants-10-01450-f002:**
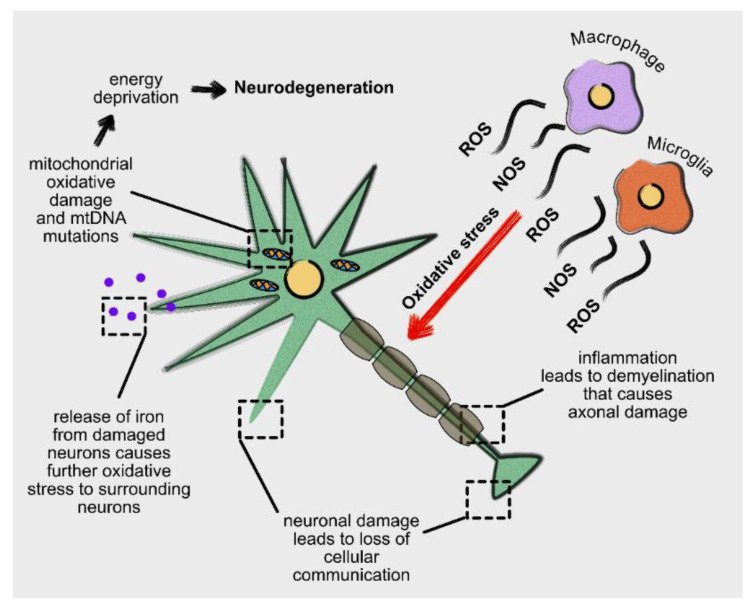
The interplay between neuroinflammation and neurodegeneration. Reactive microglia and infiltration of the nervous system by the peripheral immune cells may cause inflammation. Neuroinflammation may lead to oxidative stress, axonal damage, loss of cellular communication, mitochondrial dysfunction and eventually, may contribute to neurodegeneration.

**Figure 3 antioxidants-10-01450-f003:**
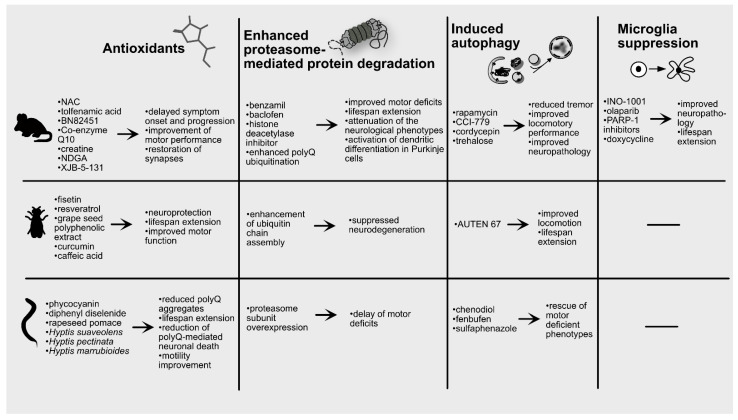
Effects of antioxidants and UPS, autophagy, microglia modulation in polyQ animal models. Various antioxidants were repeatedly shown to exert neuroprotective effects in mouse, *Drosophila* and *C. elegans* polyQ models. Similarly, genetic or compound-mediated enhancement of UPS components and autophagy results in attenuation of the symptom manifestation and/or improvement of the animal physiology. Finally, microglia suppression was found to be beneficial in mouse polyQ models.

**Table 1 antioxidants-10-01450-t001:** Genes involved in the pathogenesis of polyQ diseases along with their manner of transmission and the number of repeats that characterize them as normal or pathogenic.

PolyQ Disease	Causative Gene	Transmission	Normal/Mutant Repeat Tract Length
Spinocerebellar ataxia type 1 (SCA1)	ATXN1	Autosomal Dominant	6–35/39–83
Spinocerebellar ataxia type 2 (SCA2)	ATXN2	Autosomal Dominant	13–31/>32
Spinocerebellar ataxia type 3 (SCA3)	ATXN3	Autosomal Dominant	12–41/52–91
Spinocerebellar ataxia type 6 (SCA6)	CACNAIA	Autosomal Dominant	4–18/20–33
Spinocerebellar ataxia type 7 (SCA7)	ATXN7	Autosomal Dominant	7–27/37–460
Spinocerebellar ataxia type 17 (SCA17)	TBP	Autosomal Dominant	25–42/46–55
Dentatorubral-pallidoluysian atrophy (DRPLA)	ATN1	Autosomal Dominant	6–35/48–93
spinal and bulbar muscular atrophy (SBMA)	AR	X-linkedDominant	11–36/38–62
Huntington’s disease (HD)	HTT	Autosomal Dominant	6–35/40–121

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
