# Peer review of "Oxidative Stress and Neurodegeneration: Interconnected Processes in PolyQ Diseases"

_antioxidants, 2021, doi:10.3390/antiox10091450_

Round 1

Reviewer 1 Report

In their manuscript “Oxidative stress, neurodegeneration, and neuroinflammation: interconnected culprits in polyQ diseases”, Gkekas and colleagues compile and review the current knowledge on the disease-modulating role of oxidative stress in the context of the monogenetic polyQ disorders as observed in various cell and animal models. The authors highlight the potential interplay of oxidative stress with neurodegenerative and neuroinflammatory effects and point out potential strategies of pharmacological interventions to treat these yet incurable diseases. 

The manuscript is well written and comprises a good overview of the latest, relevant findings in the discussed context. Moreover, the key facts are nicely summarized by providing illustrative figures and a comprehensive table. 

However, several important connections and implications of oxidative stress to other mechanisms dysregulated in polyQ disorders are missing, and the overall structure of the manuscript could be optimized. 

The authors may greatly improve their manuscript by tackling the following issues:

  1. Textual:
  • The authors describe mitochondria as the main source of ROS. The process of mitophagy plays a crucial role in countering these detrimental effects, which may be further exacerbated by damaged mitochondria (see, e.g., https://doi.org/10.1016/j.mito.2020.01.002). As this knowledge is missing in the current manuscript, the authors should discuss it by adding a respective paragraph on this topic and updating their figures.  
  • As the proteasomal system, autophagy plays a significant role in polyQ disorders and is connected to oxidative stress (https://doi.org/10.1016/j.mcn.2015.03.010; https://doi.org/10.1038/cdd.2014.150). These points are missing in the manuscript, and - as mentioned for mitophagy - the authors should include the respective content in their manuscript. The effects of oxidative stress on proteasomal activity may be further demonstrated (https://doi.org/10.1016/j.redox.2017.07.005; https://doi.org/10.1155/2020/5497046). 
  • The authors may mention the role of proteolytic cleavage in disease protein toxicity in the introduction and later review the link of disease protein fragments with mitochondrial dysfunction (https://doi.org/10.2174/1381612822666161227121912). 
  • Due to the inclusion of the previous points, the discussion on pharmacological strategies might need an update (by including strategies acting on mitochondria/ autophagy in the context of oxidative stress). 
  • The detailed but also very choosy description of available animal models for polyQ diseases is slightly overloaded and may, thereby, distract from the actual focus of the review. Authors should reduce this paragraph and merge it with the facts on oxidative stress in a fashion comparable to the paragraphs on cell models of polyQ diseases.

  1. Structural:
  • Despite featuring a concise and detailed description of polyQ diseases and their characteristics, the authors may provide a summarizing table listing necessary information, such as causative gene, mode of inheritance, repeat number, etc.
  • The findings regarding oxidative stress in animal models of polyQ diseases should be moved from 4.1/ 4.2/ 4.3 and condensed in a new paragraph 3.3 oxidative stress in animal models of polyQ diseases.
  • Authors should rename and reorganize section 4, focusing specifically on the pathways impaired/ interconnected with oxidative stress [such as proteasomal system, autophagy, and microglia activation/neuroinflammation (the latter should be merged into one paragraph instead of two on HD and SCA1/SCA3 animal models)]. 
  • Section 5 is a bit redundant, far too short, and might be integrated into the previous joint paragraph of microglia activation.
  • The heading of section 5 featuring MS as a model disease for the effects discussed for polyQ disorders should be modified, stressing the presence of the connection of oxidative stress and neuroinflammation as described in this paragraph. Respectively, the title of Figure 2 should be adapted.

Author Response

Response to reviewer 1.

We would like to thank the reviewer for the productive comments. Here, are the corrections to the manuscript highlighted with a yellow color.

  1. Textual:
  • The authors describe mitochondria as the main source of ROS. The process of mitophagy plays a crucial role in countering these detrimental effects, which may be further exacerbated by damaged mitochondria (see, e.g., https://doi.org/10.1016/j.mito.2020.01.002). As this knowledge is missing in the current manuscript, the authors should discuss it by adding a respective paragraph on this topic and updating their figures.  

A separate section (4.3) describing the connection between polyQ-induced oxidative stress and mitophagy has been added in Lines 444-461.

  • As the proteasomal system, autophagy plays a significant role in polyQ disorders and is connected to oxidative stress (https://doi.org/10.1016/j.mcn.2015.03.010; https://doi.org/10.1038/cdd.2014.150). These points are missing in the manuscript, and - as mentioned for mitophagy - the authors should include the respective content in their manuscript.

A separate section (4.2) describing the connection between polyQ-induced oxidative stress and autophagy has been added in Lines 394-442.

The effects of oxidative stress on proteasomal activity may be further demonstrated (https://doi.org/10.1016/j.redox.2017.07.005; https://doi.org/10.1155/2020/5497046). 

These effects are briefly described in the new text inserted in Lines 341-345.

  • The authors may mention the role of proteolytic cleavage in disease protein toxicity in the introduction and later review the link of disease protein fragments with mitochondrial dysfunction (https://doi.org/10.2174/1381612822666161227121912). 

Short comments have been included in Lines 101-102 and 246-248

  • Due to the inclusion of the previous points, the discussion on pharmacological strategies might need an update (by including strategies acting on mitochondria/ autophagy in the context of oxidative stress). 

A relevant section (6.1.3) has been in included in Line 695-723 and Figure 3 has been updated accordingly.

  • The detailed but also very choosy description of available animal models for polyQ diseases is slightly overloaded and may, thereby, distract from the actual focus of the review. Authors should reduce this paragraph and merge it with the facts on oxidative stress in a fashion comparable to the paragraphs on cell models of polyQ diseases.

The paragraph describing the in vivo models of selected polyQ diseases has been deleted and merged with Section 3.3 in Lines 256-326

  1. Structural:
  • Despite featuring a concise and detailed description of polyQ diseases and their characteristics, the authors may provide a summarizing table listing necessary information, such as causative gene, mode of inheritance, repeat number, etc.

A relevant Table 1 has been added in Line 58

  • The findings regarding oxidative stress in animal models of polyQ diseases should be moved from 4.1/ 4.2/ 4.3 and condensed in a new paragraph 3.3 oxidative stress in animal models of polyQ diseases.

This has been done as suggested

  • Authors should rename and reorganize section 4, focusing specifically on the pathways impaired/ interconnected with oxidative stress [such as proteasomal system, autophagy, and microglia activation/neuroinflammation (the latter should be merged into one paragraph instead of two on HD and SCA1/SCA3 animal models)]. 

This has been done as suggested

  • Section 5 is a bit redundant, far too short, and might be integrated into the previous joint paragraph of microglia activation.

Section 5 has been merged with Section 6

  • The heading of section 5 featuring MS as a model disease for the effects discussed for polyQ disorders should be modified, stressing the presence of the connection of oxidative stress and neuroinflammation as described in this paragraph. Respectively, the title of Figure 2 should be adapted.

The heading of new Section 5 and the title of Figure 2 have been modified.

Reviewer 2 Report

in the manuscript authored by Ioannis Gkekas et al. authors provided an attempt revision of the literature about Neurodegenerative polyglutamine (polyQ) disorders focusing on oxidative stress and neuroinflammation.

The manuscript is well-written and structured. Here below I reported my minor comments and suggestions. 

  • I suggest authors to modify the title of the manuscript as the main focus are polyQ diseases and not oxidative stress and neuroinflammation.
  • chapter 2. oxidative stress is a general chapter. Thus I suggest authors to include also some evidence in patients affected by Alzheimer's disease and Parkinson's disease not to mislead the reader since it is a shared common feature of neurodegeneration. (Mech Ageing Dev. 2020 Dec;192:111385. doi: 10.1016/j.mad.2020.111385; Aging Cell. 2019 Dec;18(6):e13031. doi: 10.1111/acel.13031. )
  • it is not clear why authors decided to dedicate a chapter to MS (chapter 6: "The interplay between neurodegeneration and neuroinflammation; the case of MS") this could be a subchapter of chapter 5 "Neuroinflammation in polyQ diseases".
  • neuroinflammation should be better described. this description is too simplistic to describe a delicate and finely orchestrate process (Yang QQ, Zhou JW. Neuroinflammation in the central nervous system: Symphony of glial cells. Glia. 2019 Jun;67(6):1017-1035. doi: 10.1002/glia.23571. Epub 2018 Dec 11. PMID: 30548343)
  • Also title of chapter 6 should be revised.
  • english revision should be performed throughout the entire manuscript. 

Author Response

Response to reviewer 2.

We would like to thank the reviewer for the productive comments. Here, are the corrections to the manuscript highlighted with a yellow color.

  • I suggest authors to modify the title of the manuscript as the main focus are polyQ diseases and not oxidative stress and neuroinflammation.

The title has been modified in order to stress the connection between oxidative stress and pathogenesis of polyQ diseases

  • chapter 2. oxidative stress is a general chapter. Thus, I suggest authors to include also some evidence in patients affected by Alzheimer's disease and Parkinson's disease not to mislead the reader since it is a shared common feature of neurodegeneration. (Mech Ageing Dev. 2020 Dec;192:111385. doi: 10.1016/j.mad.2020.111385; Aging Cell. 2019 Dec;18(6):e13031. doi: 10.1111/acel.13031. )

A relevant section has been added in Lines 141-149.

  • it is not clear why authors decided to dedicate a chapter to MS (chapter 6: "The interplay between neurodegeneration and neuroinflammation; the case of MS") this could be a subchapter of chapter 5 "Neuroinflammation in polyQ diseases".

Section 6 has been shortened and merged with Section 5 describing the interconnection between oxidative stress and neurodegeneration using as an example the neuroinflammatory disease MS

  • neuroinflammation should be better described. This description is too simplistic to describe a delicate and finely orchestrate process (Yang QQ, Zhou JW. Neuroinflammation in the central nervous system: Symphony of glial cells. Glia. 2019 Jun;67(6):1017-1035. doi: 10.1002/glia.23571. Epub 2018 Dec 11. PMID: 30548343)

A relevant section better describing neuroinflammation has been added in Lines 539-552

  • Also title of chapter 6 should be revised.

The title of the new Section 5 has been modified

  • english revision should be performed throughout the entire manuscript.

Spell check has been performed. If the reviewers think that the manuscript needs further refinement, the final version of the manuscript will be edited accordingly.

Round 2

Reviewer 1 Report

The authors have responded to all my comments and revised their manuscript to my satisfaction. Thus, their considerably improved work can be accepted in the present form.